# Antidiabetic Medicinal Plants Used in the Eastern Cape Province of South Africa: An Updated Review

**Idowu Jonas Sagbo *** and **Ahmed A. Hussein ***

Chemistry Department, Cape Peninsula University of Technology, Symphony Road,
Bellville Campus, Bellville 7535, South Africa
* Correspondence: sagboi@cput.ac.za (I.J.S.); mohammedam@cput.ac.za (A.A.H.)

**Abstract:** Oral antidiabetic drugs are usually costly and are associated with several adverse side effects. This has led to the use of medicinal plants that are considered to have multiple therapeutic targets and are readily accessible. In the Eastern Cape province of South Africa, the number of people using medicinal plants for the management of diabetes has been climbing steadily over the past two decades due to their cultural acceptability, accessibility, affordability, efficacy, and safety claims. In this study, a review of antidiabetic medicinal plants used in the Eastern Cape province of South Africa was conducted. A comprehensive literature survey was thoroughly reviewed using several scientific databases, ethnobotanical books, theses and dissertations. About forty-eight (48) plant species were identified as being used to treat diabetes by the people of Eastern Cape province. Among the plant species, only eight (8) species have not been scientifically evaluated for their antidiabetic activities and twenty antidiabetic compounds were isolated from these medicinal plants. This review has confirmed the use and potential of the antidiabetic medicinal plants in the Eastern Cape province and identified several promising species for further scientific investigation.

**Keywords:** antidiabetic drugs; medicinal plants; diabetes mellitus; eastern cape; hyperglycaemia; hypoglycaemia

## 1. Introduction

Diabetes mellitus (DM), generally known as diabetes, is a non-communicable metabolic disease described by an abnormal increase in blood sugar (glucose) levels due to complete or relative lack of insulin secretion, with concomitant modifications in the metabolism of lipids, proteins and carbohydrates [1,2]. The chronic hyperglycaemic status in diabetics yields an increased risk of complications due to the long-term damage and dysfunction of several organs such as the eyes, kidneys, heart, and nerves. Symptoms of clear hyperglycaemia include weight loss, blurred vision, polyuria (excessive passage of urine), polydipsia (excessive thirst), and susceptibility to infections, whereas long-term complications may include retinopathy (damage to the retina of the eyes) with possible loss of vision, nephropathy (deterioration of kidney function), and cardiovascular complications such as high blood pressure (hypertension).

Some specific cases of diabetes have been documented, but the vast majority of cases fall into two broad classes, namely, Type 1 diabetes and Type II diabetes. Type 1 diabetes or insulin-dependent diabetes is caused by a relative or absolute deficiency of insulin secretion, commonly due to cell-mediated autoimmune destruction of beta cells of the pancreas, which may have a genetic predisposition [3]. Type 1 diabetes accounts for 5–10% of patients with diabetes. On the other hand, Type 2 diabetes or non-insulin dependent diabetes, or adult-onset diabetes accounts for about 90–95% of diabetic patients and is linked with reduced insulin sensitivity, also known as insulin resistance and/or impaired insulin secretion. The pathogenesis of type 2 diabetes is more flexible, though the autoimmune destruction of beta cells does not occur [4].

## 1.1. Prevalence of Diabetes

Over the past two decades, the number of people diagnosed with diabetes has reached an unprecedented high and a further increase is expected. The International diabetes federation (IDF) reported that there are 463 million (20–79 years) people globally suffering from this life-threatening disease and this figure is anticipated to increase to 700 million by 2040 [5]. The World Health Organization (WHO) also indicated that if the existing trend lingers, diabetes will be the second highest killer by 2040 unless robust and rigorous actions are made by individuals, communities and governments [6]. These are part of an awareness campaign on the burden of diabetes and the urgency to intensify prevention and control activities. Globally, China has the highest number of people living with diabetes, followed by India and the United States [5]. Asia and Africa have been identified as areas with high diabetic populations which could increase beyond projected levels if urgent attention is not given [5]. In South Africa, there were about 4.5 million cases of diabetes in 2019, with occurrences in the same period placed at 12.7% among adults 20–79 years old [5]. Inadequate promising therapy to cure diabetes could be accountable for the predicted figure in this part of the world.

## 1.2. Oral Antidiabetic Drugs and Their Limitations in the Treatment of Diabetes Mellitus

Currently, existing oral hypoglycaemic drugs are categorized into different groups according to their mechanisms of action (Figure 1). The list includes, but is not limited to sulfonylureas, meglitinides, biguanides, thiazolidinediones, alpha-glucosidase inhibitors and dipeptidyl peptidase 4 (DPP-IV) inhibitors.

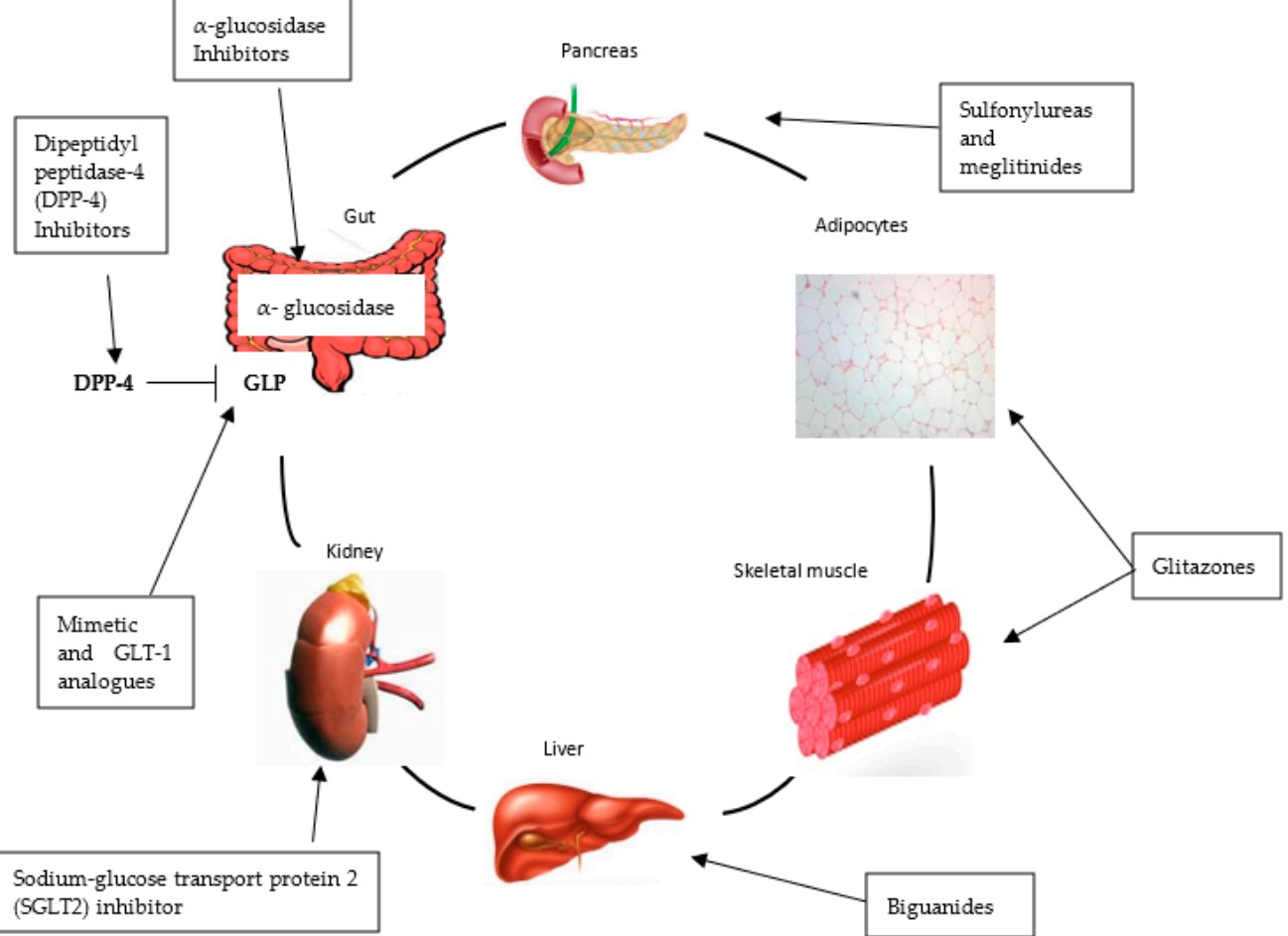

**Figure 1.** Target tissue of current antidiabetic drugs.

The sulfonylureas act by stimulating insulin release from the pancreatic beta cells [7]. They bind to the SUR-1 (sulfonylurea receptor-1), expressed on the pancreatic beta cell membranes, thereby inhibiting the efflux of potassium ions through the channels that cause depolarization [7]. This depolarization causes the opening of voltage-gated calcium channels, leading to the increased influx of calcium, and this rise in intracellular calcium stimulates insulin release [8]. However, it has been proven that these antidiabetic drugs do not reduce the long-term complication of diabetes and may also enhance appetite hereafter, resulting in weight gain [9]. Examples of drugs in this class are tolbutamide, tolazamide, acetohexamide chlorpropamide, glyburide, glipizide, glibenclamide and glimepiride.

Meglitinides are another class of oral antidiabetic drugs. These drugs reduce blood sugar levels, thereby increasing insulin release from the beta-cell pancreatic [10]. This is achieved by modulating beta cells to secrete insulin by controlling the efflux of potassium through potassium channels. Meglitinides do not have a direct effect on insulin exocytosis as it is in sulfonylureas [11]. This class of antidiabetic drug is taken by diabetic patients shortly before a meal to increase the insulin response to each meal [12]. If a meal is skipped the medication is also skipped. The reported side effects associated with meglitinides include weight gain and hypoglycaemia [13]. Typical examples of the antidiabetic drugs in this class are repaglinide and nateglinide.

The biguanides, another class of oral antidiabetic drugs, work by enhancing insulin sensitivity in peripheral tissues through the alteration of post-receptor signalling in the insulin signalling pathway. Their effects on hepatic tissue result in reduced hepatic glucose production through a reduction in gluconeogenesis and glycogenolysis [14,15]. The best example of this class is metformin. However, metformin has been described to have certain adverse side effects such as heart failure, hepatic impairment, gastrointestinal disturbances and renal impairment [7,16].

The thiazolidinediones are another class of oral antidiabetic drugs that mediate their function by binding to the PPARγ (peroxisome proliferator-activated receptor gamma) mainly expressed in adipocytes. Binding to PPARγ stimulates interaction with the retinoid X receptor, which hetero-dimerizes and activates genes that play a significant role in lipid and carbohydrate metabolism [17]. They help to improve muscle and fat sensitivity to insulin and, to a lesser extent, reduce hepatic glucose production. However, thiazolidine-diones have been described to be linked with the pathophysiology of fluid-retention and weight increase. The complete effects of fluid retention include oedema, heart failure, liver toxicity and anaemia [18]. Examples of drugs in this class include troglitazone, rosiglitazone and pioglitazone.

The alpha-glucosidase inhibitor class of oral antidiabetic drugs is occasionally called "starch blockers". They block the action of intestinal enzymes that break down carbohydrates in the small intestine, thus slowing down the absorption of ingested carbohydrates and decreasing post-prandial hyperglycaemia in diabetic patients [19]. However, one of the biggest drawbacks of these oral antidiabetic drugs is their side effects. The prominent side effects include nausea, flatulence, diarrhoea, bloating and abdominal pains [11]. Acarbose and miglitol are examples of this type of oral antidiabetic drug.

The dipeptidyl peptidase 4 (DPP-IV) inhibitors are a new class of oral antidiabetic drugs. They attenuate incretin degradation, thus increasing the half-lives of incretin and enhancing the stimulation of pancreatic insulin secretion and beta cell growth [20]. Incretin hormones (glucose-dependent insulinotropic peptide (GIP) and glucagon-like peptide (GLP-1)) contribute meaningfully to glucose-dependent insulin secretion by increasing beta cell mass and decreasing glucagon secretion. These gut hormones (GIP and GLP-1) are highly sensitive to degradation by DPP-IV, a serine protease that cleaves polypeptides containing proline and alanine residues at the penultimate N-terminal position and thus decreases the effectiveness of the hormones. Adverse side effects of DPP-IV inhibitors include headache, runny nose, diarrhea, nausea, stomach pain, and sore throat. The best examples of this class are alogliptin, linagliptin, sitagliptin and saxagliptin.

*1.3. Plants as an Alternative Source of Antidiabetic Agents*

The use of herbs to treat diabetes mellitus has been highly acceptable as part of medical intervention. This is due to their efficacy, probable fewer side effects and lesser costs [21]. In developing countries, medicinal plants are used to treat diabetes to overwhelm the problem of the cost of western medicines to the populace [22]. These medicinal plants have been indicated to contain several active chemical constituents that are accountable to treat diabetes [22]. Plants have always served as a good source of drugs with some of the existing antidiabetic drugs being acquired directly or indirectly from them [23–25]. For example, metformin, the favoured first-line oral antidiabetic drug, emanates from a derivate of French lilac (*Galega officinalis*) [26].

There have been several review reports on plants with probable antidiabetic activity from different parts of the world in the literature [27–29]. In South Africa, several ethnobotanical surveys have shown numerous plants used traditionally to treat diabetes [30–32]. Some plants with antidiabetic effects from South Africa are available in the form of herbal supplements, possessing the NAPPI code, a unique coding identifier for medicine, surgical products and medical procedures. For example, Probetix (Nappi code: 711050-001) is an herbal supplement made from the *Sutherlandia frutescens* extract (main active component: pinitol). This supplement has been reported to cause a reversal of insulin resistance and reduce intestinal glucose uptake [33]. Manna DFM43 (Nappi code: 705846-001) developed from the pods of *Prosopis glandulosa* (main active component: galactomannan) is known to slow down glucose absorption and also reduce the glycaemic index of foods. Despite the enormous use of these plants in the treatment of life-threatening diseases such as diabetes, there are still several plants that remain to be recorded.

For many years, people from the Eastern Cape province, one of the poorest provinces in South Africa, combined with the highest provincial unemployment rate (55%), relied heavily on the use of medicinal plants to treat diabetes. This is mainly due to their cultural acceptability, accessibility, inexpensiveness, efficacy, and safety claims [34,35]. The Xhosas are the major inhabitants of the Eastern Cape province. Despite the western influence, the Xhosa people still believe in the effectiveness of medicinal plants and prefer to use these plants currently. This review aims to highlight the antidiabetic medicinal plants used in the Eastern Cape province of South Africa with the view of preventing the loss of vital traditional knowledge of plants used to treat diabetes. This review is expected to identify the existing knowledge gaps and serves as an important baseline for future research on scientifically underexploited plant species.

**2. Methods**

A comprehensive literature survey was thoroughly conducted from January 2021 to August 2022. A report about the antidiabetic medicinal plants used traditionally in the management of diabetes in the Eastern Cape province was thoroughly retrieved from various scientific databases such as Google Scholar, Science Direct, PubMed, Medline, Scopus and Web of Science. In addition, theses, dissertations, and ethnobotanical books were also retrieved from various university libraries. The keywords and terms used to obtain relevant articles or information were the "scientific name of the plants", "antidiabetic", "hypoglycaemia", mode of action, "diabetes" and "ethnopharmacology".

**3. Results and Discussion**

*3.1. Plants Used in the Eastern Cape Province with Antidiabetic Potentials*

Forty-eight plant species were identified as being used to treat diabetes by the people of Eastern Cape province (Table 1), although some of these plants have been scientifically examined for their antidiabetic activity, but eight of these plant species are yet to be scientifically investigated (Table 2). For the purpose of this review, a comprehensive description of the traditional usage, antidiabetic mechanism of action and the active molecules (Figure 2) of some of the plants used in the Eastern Cape province for the treatment of diabetes are as follows:

**Figure 2.** Antidiabetic compounds isolated from Eastern Cape antidiabetic medicinal plants. The numbers 1–20 correspond to the compounds reported in Table 2.

### 3.2. Albuca setosa Jacq.

*Albuca setosa* (Figure 3) (Hyacinthaceae family) is a plant that has narrow leaves that become broader at the base. It is a perennial, very hardy, evergreen plant that grows from 0.15 m to 0.5 m [36]. The plant is dispersed in most parts of South African provinces, except for the extreme South-Western Cape province. Some of these provinces include the Eastern Cape, Limpopo, Free State Northern Cape, KwaZulu-Natal, Gauteng and Mpumalanga. Traditionally, the fresh corms of the plant are crushed, boiled and taken orally to treat diabetes [37]. Additionally, the plant is used for the treatment of wounds, articulation complications, rheumatoid arthritis, digestive disorder and venereal diseases in human beings [38]. The aqueous extract of *A. setosa* corms showed high glucose utilization in cultured L6 muscle and 3T3-L1 cells [37]. The extract also exhibited weak inhibition against alpha-amylase activity but strongly inhibit alpha glucosidase with $IC_{50}$ value of 7.725 mg/mL [37]. This antidiabetic activity of *A. setosa* has been attributed to its main bioactive compounds such as phenols and flavonoids [37].

### 3.3. Artemisia afra Jacq. ex Willd.

*Artemisia afra* is one of the most frequently used medicinal plants in South Africa. The plant belongs to the Asteraceae family. It grows in bushy, slightly untidy clumps, commonly with high stems up to 2 m in height, but occasionally as low as 0.6 m [39]. The plant is widespread in all South African provinces, except for the Northern Cape province. It is also found in Swaziland, Lesotho, and northwards into tropical Africa [31]. The leaves or roots of *A. afra* are used mostly as an infusion and then taken orally by the people of the Eastern Cape to treat diabetes [31]. Studies investigated by Afolayan and Sumonu [40] revealed that *A. afra* exhibited a strong ability to reverse diabetic oxidative stress in streptozotocin-induced diabetic rats. One report also indicated that *A. afra* extract showed a significant reduction in blood glucose levels with the greatest reduction seen in the 200 mg/kg concentration, which is almost the same effect as the standard positive control, glibenclamide, used in the study [41]. It has also been reported that acetone extract of *A. afra* showed weak inhibition of alpha-glucosidase [41]. In a separate study, the active antidiabetic molecules reported from *A. afra* are polyphenols, flavonoids, sterols, alkaloids and terpenoids [42]. Literature surveys revealed no reported antidiabetic isolated compound from this plant.

### 3.4. Brachylaena discolor DC.

*Brachylaena discolor* (Asteraceae family) is an evergreen shrub with a height of 4 to 10 m. It is found in coastal forests, bushes and on the margins of the evergreen forest in the Eastern Cape province of South Africa to Mozambique. The leaf infusion of the plant is used to treat diabetes in the Eastern Cape province [31]. In addition, the infusion is also used as a tonic to treat intestinal parasites and chest pain. A literature report revealed that the aqueous extract of *B. discolor* stimulated glucose uptake in 3T3-L1 and C2C12 muscle cells [43]. Mellem et al. [44] also reported that aqueous extract of *B. discolor* showed a strong inhibition against alpha-glucosidase. In the vivo studies, the methanol extract of *B. discolor* exhibited a marked decline in the blood glucose level of a diabetic rat at the tested concentration [45]. The literature survey showed no report on the antidiabetic activity of isolated compounds. Therefore, additional studies are required to elucidate the antidiabetic molecules present in the plant.

### 3.5. Bulbine frutescens (L.) Willd.

*Bulbine frutescens* is a common, waterwise garden plant that belongs to the Xanthorrhoeaceae family. It is a fast-growing plant with linear green leaves in opposite rows and grasping the stems at the base. The plant occurs extensively throughout parts of the Western Cape, Eastern Cape and Northern Cape provinces of South Africa. The root infusion of the plant is taken orally to treat diabetes by the people of the Eastern Cape [34]. An infusion of fresh leaves in a cup of boiling water is taken for coughs, arthritis and colds [46]. A report by van Huyssteen et al. [47] showed that whole plant aqueous extract of *B. frutescens* increased glucose uptake in Chang liver cells. This effect was greater than the standard positive control, metformin, used in the study. Nevertheless, there are a lack of scientific data on the antidiabetic in vivo studies of *B. frutescens* in the literature. No reports on antidiabetic isolated compounds from *B. frustescens* have yet been carried out.

### 3.6. Catha edulis (Vahl) Forrsk. ex Endl.

*Catha edulis* belongs to the Celastraceae family, normally known as the spike thorn family. The plant is very attractive and is found in woodlands and on rocky outcrops. It is distributed in the Eastern Cape province, mostly from the mist belt, moving inland. The fresh leaves of the plant are chewed to treat diabetes [48]. The in vitro studies revealed that *C. edulis* dichloromethane/methanol (1:1) extract stimulated glucose uptake in C2C12 and 3T3-L1 fat cells. In animal studies, the aqueous extract of the plant exhibited significant hypoglycaemic and weight reduction effects in normal streptozocin-induced diabetic rats [48]. Piero et al. [49] also reported that the aqueous extract of *C. edulis* effectively lowered blood

glucose levels to normal in alloxan-induced diabetic rats compared to insulin used in the study. This antidiabetic effect of the plant has been ascribed to the presence of chemical components, but the elucidation or isolation of the antidiabetic molecules present in the plant is still required.

### 3.7. Conyza scabrida DC.

*Conyza scabrida* is a plant that consists of fresh or dried leaves and small stems. The plant belongs the Asteraceae family and is found in streamside and forest margins of the Western and Eastern Cape provinces of South Africa. The fresh leaves of the plant are taken orally as an infusion to treat diabetes [50]. The plant is also used as an external application to treat sores and inflammation [51]. The literature survey revealed no scientific investigation on its antidiabetic properties.

### 3.8. Dianthus thunbergii S.S. Hooper

*Dianthus thunbergii* is referred to as "wild pink", due to the color of its flowers, belonging to the genus *Dianthus*, family Caryophyllaceae. The plant is 30 cm high, and its flowers are pale pink with bracts approximately 4 cm long with fine grey-blue leaves at the base. It is found in South Africa and occurs often in the Eastern part of the country [52]. The extract from the freshly crushed roots of the plant is reportedly used against diabetes [30]. The ethanol and aqueous root extract of *D. thunbergii* have been reported to exhibit moderate glucose uptake in L6 muscle cells in vitro [53], but the scientific efficacy of the plant in in vivo studies has not been validated. Additionally, isolated antidiabetic compounds from *D. thunbergia* are yet to be reported in the literature.

### 3.9. Euclea undulata Thunb.

*Euclea undulata* is a dense, erect, grassy, perennial dioecious shrub belonging to the family Ebenaceae. It is one of the most common small trees across the enormous subtropical and central interior regions of Southern Africa. Its flowers are very small and pale in ancillary racemes up to 2 cm long. The plant is widespread in Southern African countries. In South Africa, the plant occurs extensively on rocky slopes throughout all provinces [31]. Traditionally, an infusion from the ground root bark of the plant is drunk as a tea to treat diabetes by Eastern Cape people [54]. The root bark acetone extract of the plant has been described to increase glucose uptake in C2C12, Chang liver and 3T3-L1 cells exhibited a strong effect against alpha-glucosidase [7]. It has been reported that the root bark acetone extract of *E. undulata* also decreased fasting blood glucose levels, elevated cholesterol and triglyceride levels to near normal in streptozotocin–nicotinamide-induced type 2 diabetic rats without any weight gain at a dose of 100 mg/kg body weight [55]. Several antidiabetic compounds isolated from this plant have also been reported for their antidiabetic activity [56]. Epicatechin isolated from *E. undulata* was reported to reduce blood glucose levels in C2C12 cells [56]. Another compound, α-amyrin-3O-β-(5-hydroxy) ferulic acid isolated from *E. undulata* showed strong inhibition against alpha-glucosidase activity with an $IC_{50}$ value of 4.79 [56].

### 3.10. Hypoxis colchicifolia Bak.

*Hypoxis colchicifolia* (Hypoxidaceae family) is the second most vital *Hypoxis* medicinal species with marketable value in South Africa. It grows in grasslands between 25 and 60 cm in height. The plant possesses large underground tubers which enable it to subsist the constant grass fires common to this vegetation type. *H. colchicifolia* is found on sandy or poor soil in grassland across the provinces of Eastern Cape and KwaZulu-Natal [57]. In traditional medicine, the fresh corms of the plant are milled, boiled in water and then taken orally to treat diabetes. The in vitro antidiabetic studies conducted by Cumbe [58] revealed that the methanol extract of the plant exhibited moderate glucose utilization in C2C12 muscle cells and Chang liver cells, but there is no scientific studies supporting this claim in the animal model. Active antidiabetic molecules isolated from *H. colchicifolia* include

hypoxoside and bisphenol A diglycidyl ether [58]. It should be noted that bisphenol A diglycidyl ether is an environmental impurity. This antidiabetic activity of *H. colchicifolia* has been attributed to its main bioactive compounds.

### 3.11. Leonotis leonorus (L.) R.Br.

*Leonotis leonurus* (Lamiaceae family) is referred to as a lion's tail. It is a fast-growing, soft-woody, strong shrub that grows up to 2–3 m high and 1.5 m wide. It is very common and widespread across all South African provinces, most especially the Eastern Cape, where it grows between rocks and in grassland [59]. A decoction from the whole part of the plant is taken orally for the treatment of diabetes by the people of the Eastern Cape [30]. Very little has been done to establish the in vitro antidiabetic properties of this plant. The aqueous extract of *L. leonurus* has been reported to exhibit hypoglycaemic effects in streptozotocin-induced diabetic rats, thereby decreasing blood glucose levels as well as low-density lipoprotein [60]. Another study conducted by Odei-Addo et al. [61] revealed that the extract Nanostructured lipid carriers (NLCs) formulation improved glucose uptake in liver cells. The antidiabetic activity of this plant has been ascribed to various polyphenolics, diterpenoids, and flavonoids present in the extract [62]. Marrubin, an active antidiabetic compound, has been isolated from the organic extracts of *L. leonurus* [63].

### 3.12. Momordica foetida Schumach.

*Momordica foetida* (Cucurbitaceae family) is an herbaceous, climbing, perennial herb producing annual stems up to 4.5 m. The plant has dark green flecks when young and woody when old. It is mostly found in the Eastern Cape province of South Africa. Traditionally, the whole plant is apparently used to treat diabetes in the Eastern Cape [30]. In addition, the juice of crushed leaves from the plant is used to relieve intestinal disorders [30]. Van der venter et al. [43] reported the in vitro antidiabetic activity of *M. foetida* dichloromethane/methanol (1:1) extract, thereby showing an increase in glucose uptake in C2C12 muscle cells. Akinwumi [64] also reported that the effects of the methanol extract of *M. foetida* had low inhibitory activity against alpha-amylase and alpha-glucosidase activities. Not much has been done to validate the effect of the plant in animal model studies. However, foetidin, an isolated compound from *M. foetida*, has been reported to decrease blood glucose levels in fasting and alloxan-treated rats [65].

### 3.13. Psidium guajava L.

*Psidium guajava* (Myrtaceae family) is a well-known tropical tree rich in fruit. It is a perennial shrub-like tree that ranges in height from 6 to 25 ft [31]. The plant has a wide-ranging dispersed network of branches. The leaves of the plant are extensive and green in colour, with noticeable veins to cure wounds, ulcers and toothache [66]. *P. guajava* is widely dispersed in the provinces of KwaZulu-Natal, Eastern Cape, Mpumalanga and Limpopo [31]. In the Eastern Cape, the warm water extracted from the dried leaves of the plant is taken orally to treat diabetes [67] The leaves of the plant are also used for other ailments such as boils, wounds, and coughs [68]. It has been reported that *P. guajava* ethanol extract demonstrates a considerable decrease in blood glucose levels in alloxan-induced diabetic rats at an oral dose of 250 mg/kg [69]. Another study reported by Shukla and Dubey [70] revealed that the aqueous and ethanolic extracts of *P. guajava* produced blood glucose homeostasis but also reversed metabolic and pathologic changes in pancreatic islets. Tella et al. [71] also reported that the treatment of diabetic animals with the *P. guajava* extract ameliorated damage to the pancreatic islets and enhanced the lowering of blood glucose. This antidiabetic activity of the plant has also been confirmed in in vitro studies. Another report by van de venter et al. [43] revealed that the leaves of dichloromethane/methanol (1:1) extract of *P. guajava* increased glucose uptake in C2C12 and 3T3-L1 cells. The acetone extract of the plant also showed good inhibition against alpha-amylase and alpha-glucosidase activities [41]. The aqueous leaf extract of *P. guajava* was also reported to exhibit excellent inhibitory activity against alpha-glucosidase with an $IC_{50}$

value of 5.6 mg/mL [72]. The major active antidiabetic compound isolated from *P. guajava* includes arachidic acid (1), β -sitosterolxylopyranoside (2), lanost-7-en-3β-ol-26-oic acid (3) lanost-7-en-3β, 12β-diol-26-oic acid (4), lanost-7-en-3β, 12β, 29-triol-26-oic acid (5), lanost-cis-1,7,23-trien-3β, 12β, 18, 22α-tetraol-26-oic acid (6), lanosteryl-3β-O-D-xylopyranosyl-2′-*p*-benzaldehyde (7) and lanost-7-en-3β-ol-26-oic acid-3β-D-glucopyranoside (8) [66]. The isolated compounds 3,4,5 and 8 have been reported to show strong antidiabetic activity against streptozotocin-induced diabetic rats [66].

### 3.14. Solanum aculeastrum Dunal subsp. aculeastrum

*Solanum aculeastrum* (Solanaceae family) is a small tree of approximately 1–5 m high. The leaves of the plant are shortly petiolate, ovate, up to 150 × 130 mm. The plant is found naturally in grassland, woodland and in forest margins across some South African provinces, particularly Eastern Cape and Western Cape provinces. Traditionally, the decoction from the root of the plant is taken orally to treat diabetes in the Eastern Cape [30]. However, there is still no scientific report of its antidiabetic activity. The literature survey also showed no reported antidiabetic isolated compounds from *S. aculeastrum*.

### 3.15. Sutherlandia frutescens (L.) R.Br.

*Sutherlandia frutescens* (Fabaceae family) is a small plant around 1.2 m in height. It is a fast-growing and drought-tolerant plant that prefers full sun. The compound leaves of the plant are greyish green in colour. It is broadly dispersed in the dry areas of the Western, Eastern, Northern Cape provinces, frequently in distressed places [31]. The plant is used by indigenous communities throughout South Africa. In the Eastern Cape province, the leaf infusion of the plant is used to treat diabetes [50]. It is also used for the treatment of gastric ailments, cancer, and gynaecological complications [31]. The literature survey showed various in vitro antidiabetic studies of the plant. William et al. [73] stated that the aqueous extract of *S. frutescens* prevented insulin resistance (a precursor of type II diabetes) in Chang liver cells. Another report by Elliot, [74] also revealed that the aqueous extract of the plant stimulated insulin secretion from INS-1 cells. MacKenzie et al. [75] also showed that the plant extract enhanced glucose uptake in 3T3-L1 adipocytes. To confirm its antidiabetic activity in the animal model, Chadwink et al. [33] revealed that the *S. frutescens* extract exhibited a strong capacity to normalize insulin levels, increased glucose uptake in peripheral tissues, and also subdue intestinal glucose uptake with no significant weight gain in pre-diabetic rats. The active antidiabetic compounds isolated from *S. frutescens* include flavonoids, pinitol, saponins and triterpenoid [76]. Among the antidiabetic compounds, pinitol, a well-known antidiabetic agent, has been reported to have the same properties as insulin [77]. Therefore, the presence of pinitol explains the antidiabetic use of *S. frutescens*.

### 3.16. Tarchonanthus camphoratus L.

*Tarchonanthus camphoratus* is a semi-deciduous small tree with a height of 2–9 m. *T. camphoratus* belongs to the Asteraceae family. The leaves of the plant are thin, with entire finely toothed margins. *T. camphoratus* is widely spread in a range of habitats, such as thickets of bushveld, grassland and forests across Southern African countries. In South Africa, the plant is widely dispersed in the Eastern Cape, Gauteng, Free State and Northern Cape provinces. In traditional medicine, the fresh leaves of this plant are taken orally as an infusion to treat diabetes in the Eastern Cape [30]. The literature report indicated that the aqueous and ethanol extracts of *T. camphoratus* exhibited high glucose uptake in C2C12 muscle cells [47]. However, there are no scientific records of its antidiabetic activity in an animal model. Additionally, no isolated antidiabetic molecule has been reported from *T. camphoratus*.

### 3.17. Vernonia oligocephala Sch. Bip.

*Vernonia oligocephala* (Table 1) is an upright, perennial, herbaceous plant that belongs to the Asteraceae family. It measures up to 1 m in height and its stems grow from a

woody rootstock. It is widely spread throughout the grassland regions in several South African provinces, including the Eastern Cape [31,57]. The infusion of the leaves of this plant is taken orally to treat diabetes [34]. The infusion of the leaves is also used to cure rheumatism and dysentery [68]. The methanol extract of *V. oligocephala* has been reported to show weak inhibition against alpha amylase inhibition [78]. Additionally, oligocephalate, a compound isolated from *V. oligocephala*, was reported to exhibit strong inhibition against alpha glucosidase enzymes, with an $IC_{50}$ value of 18.5 μM. [79].

**Table 1.** List of medicinal plants used to treat diabetes in the Eastern Cape province.

| S/N | Scientific Name | Local Name (Xhosa) | Family | Part Used | Method of Preparation | Reference |
|---|---|---|---|---|---|---|
| 1 | *Albuca setosa* Jacq. | Ingwe beba | Hyacinthaceae | Corms | Fresh corms are milled, boiled and taken orally. | [30] |
| 2 | *Allium sativum* L. fam. | Ikoronofile | Alliaceae | Whole plants | The fresh plant is crushed, boiled and the infusion is taken orally. | [30] |
| 3 | *Artemisia afra* Jacq. ex Willd. | Umhlonyane | Asteraceae | Leaves, roots | The leaves or roots are boiled, then the infusion is mixed with sugar to reduce the bitterness. | [34] |
| 4 | *Aloe ferox* Mill | Ikhala- lasekoloni | Aloaceae | Leaves | The liquid from the leaves is boiled to powder, submerge in water and taken orally. | [30] |
| 5 | *Anacampseros ustulata* E.Mey. ex Sond. | Igwele | Portulaceae | Corms | Fresh corms are crushed, boiled and taken orally. | [30] |
| 6 | *Bridelia micrantha* (Hochst.) Baill. | umhlalamakwaba, | Phyllanthaceae | Stem bark. | Unspecified | [79] |
| 7 | *Brachylaena discolor* DC. | UmPhahla | Asteraceae | Leaves | Leaves of the plant are boiled, and the infusion is taken orally | [80] |
| 8 | *Brachylaena elliptica* (Thunb.) DC. | isiduti | Asteraceae | Leaves | An infusion is made from fresh leaves serves as a gargle and mouthwash | [31,81] |
| 9 | *Brachylaena ilicifolia* (Lam.) E. Phillips & Schweick. | Umgqeba. | Asteraceae | Leaves | An infusion is prepared from fresh leaves serves as a gargle | [31] |
| 10 | *Bulbine abyssinica* A.Rich | Uyakayakana | Asphodelaceae | whole plant | The whole plant parts are crushed, boiled and the infusion is taken orally. | [30] |
| 11 | *Bulbine frutescens* L. (Willd.) | Ibhucu | Xanthorrhoeaceae | Roots | The infusion is prepared from freshly boiled roots and taken orally. | [34] |
| 12 | *Bulbine natalensis* Baker | Ibhucu | Asphodelaceae | Roots | An infusion is prepared from boiled fresh roots and taken orally. | [30] |
| 13 | *Cannabis sativa* L. | Umya | Cannabaceae | Whole plants | Unspecified | [82] |
| 14 | *Carpobrotus edulis* (L.) L. Bolus | unomatyumtyum, | Mesembryanthemaceae | Leaves | The leaf juice and leaf palp are taken orally. | [31] |
| 15 | *Catha edulis* (Vahl) Forrsk. ex Endl. | iqgwaka | Celastraceae | Leaves | The leaves are chewed | [48] |
| 16 | *Catharanthus roseus* (L.) G. Don. | Isisushlungu | Apocynaceae | Leaves | The infusion is prepared from boiled leaves and is taken orally. | [34] |
| 17 | *Chilianthus olearaceus* Burch. | Umgeba | Buddlejaceae | Leaves and twigs | The fresh leaves or twigs are taken orally as an infusion. | [34] |

**Table 1.** *Cont.*

| S/N | Scientific Name | Local Name (Xhosa) | Family | Part Used | Method of Preparation | Reference |
|---|---|---|---|---|---|---|
| 18 | *Chironia baccifera* L. | NA | Gentianaceae | Rhizomes | The fresh or dry rhizomes are chewed. | [83] |
| 19 | *Cissampelos capensis* L.f. | Umayisake | Menispermaceae | Roots | The fresh corms are milled, boiled and taken orally. | [30] |
| 20 | *Conyza scabrida* DC. | Isavu | Asteraceae | Leaves | The fresh leaves are taken orally as an infusion. | [30] |
| 21 | *Dianthus thunbergii* Hooper | Ungcana, | Caryophyllaceae | Roots | The extract from the freshly crushed roots is taken orally. | [30] |
| 22 | *Euclea natalensis* A.DC. | umKhasa | Ebenaceae | Roots | The roots are used to make a decoction and then taken orally | [84] |
| 23 | *Euclea undulata* Thunb. | Umgwali | Ebenaceae | Roots, bark | An aqueous infusion from the ground root bark is usually drunk as tea. | [54] |
| 24 | *Helichrysum gymnocomum* | Impepho | Asteraceae | leaves | A decoction is prepared from boiled leaves and taken orally. | [30] |
| 25 | *Helichrysum nudifolium* (L). Less. | Ichocholo | Asteraceae | Leaves and roots | Fresh leaves or roots are boiled, then taken orally. | [34] |
| 26 | *Helichrysum odoratissimum* L. | Imphepho | Asteraceae | Whole plant | An infusion from the whole plant is milled, boiled and then taken orally. | [34] |
| 27 | *Heliichrysum petiolare* Hilliard & B.L. | Imphepho | Asteraceae | Whole plant | The whole plant is crushed, boiled and the concentrated solution is taken orally. | [34] |
| 28 | *Heteromorpha arborescens* (Spreng.) Cham. | Umbangandlala | Apiaceae | Leaves and roots | The infusion is prepared from boiled leaves or roots and taken orally | [34] |
| 29 | *Hypoxis argentae* Harv Ex Baker | Ixalanxa, | Hypoxidaceae | Corms | The corms are boiled in water and then taken orally | [30] |
| 30 | *Hypoxis colchicifolia* Bak. | Inongwe | Hypoxidaceae | Corms | Fresh corms are crushed, boiled in water and then taken orally. | [34] |
| 31 | *Hypoxis hemerocallidea* Fisch. and C. A | Inongwe | Hypoxidaceae | Corms | Fresh corms are crushed, boiled in water and then taken orally. | [34] |
| 32 | *Lauridia tetragonia* (L.f.) R.H. Archer | Umdlavuza | Celastraceae | Barks | The Infusion from the powdered bark is taken orally | [30] |
| 33 | *Leonotis leonorus* (L.) R.Br. | Unfincafincane | Lamiaceae | Whole plants | A whole plant is milled, boiled, mixed with coke and half a cup of decoction taken orally. | [30] |
| 34 | *Momordica balsamina* L. | NA | Cucurbitaceae | Fruits, Stem and flowers | The Infusion from fresh young fruit and taken orally | [31,83] |
| 35 | *Momordica foetida* Schumach. | NA | Cucurbitaceae | Whole plants | Unspecified | [82] |
| 36 | *Ornithogalum longibracteatum* (Jacq) | Ingwe beba | Hyacinthaceae | Bulb | The fresh bulb soaked in water and the concoction taken orally | [37] |
| 37 | *Psidium guajava* L. | NA | Myrtaceae | Leaves | The warm water extract from the dried leaves is taken orally | [67] |
| 38 | *Ruta graveolens* L. | iyeza lomoya | Rutaceae | Leaves | Unspecified | [43] |
| 39 | *Sclerocarya birrea* (A. Rich.) Hochst. subsp. caffra (Sond.) Kokwaro | NA | Anacardiaceae | Leaves | The decoctions or infusions from leaves taken orally | [85] |

**Table 1.** *Cont.*

| S/N | Scientific Name | Local Name (Xhosa) | Family | Part Used | Method of Preparation | Reference |
|---|---|---|---|---|---|---|
| 40 | *Solanum aculeastrum* Dunal subsp. aculeastrum | umthuma, itunga | Solanaceae | Roots | The decoction is prepared from fresh crushed roots and then taken orally | [30] |
| 41 | *Strychnos henningsii* Gilg | Umnonono | Loganiaceae | Barks | The barks are crushed to powder and half of a cup of the decoction is taken orally | [30] |
| 42 | *Sutherlandia frutescens* L. | umnwele | Fabaceae | leaves | An infusion of the leaves is taken orally | [50] |
| 43 | *Tarchonanthus camphoratus* L. | Umgqemba | Asteraceae | leaves | The fresh leaves are taken orally as an infusion | [30] |
| 44 | *Tulbaghia alliacea* L. | Umwelela | Alliaceae | Roots | The fresh roots are crushed, boiled and three quarters of a cup of decoction is taken orally | [47] |
| 45 | *Tulbaghia violacea*. Harv. | utswelane | Alliaceae | leaves | Unspecified | [47] |
| 46 | *Vernonia amygdalina* DeL. | Umhlunguhlungu | Asteraceae | Leaves | Powdered from the fresh leaves are soaked in water and the solution is taken orally. | [34] |
| 47 | *Vernonia oligocephala* Sch. Bip. | Umhlunguhlungu | Asteraceae | leaves, roots or twigs | The infusion prepared from fresh leaves, roots or twigs is taken orally. | [34] |
| 48 | *Vinca major* L. | Iflawa | Apocynaceae | Leaves, roots, stem | Unspecified | [43] |

**Table 2.** Plant used traditionally in the Eastern Cape with reported antidiabetic activity.

| S/N | Scientific Name | Plant Part Used | Solvent/Extract Used | Antidiabetic Isolated Compounds | Antidiabetic Mechanism of Action | References |
|---|---|---|---|---|---|---|
| 1 | *A. setosa* | Bulb | Aqueous and acetone | * | High glucose utilization in cell lines and weak inhibition against carbohydrate digesting enzymes | [37] |
| 2 | *A. sativum* | Corms | Ethanol | * | Inhibition against alpha-amylase | [86] |
| 3 | *A. afra* | Leaves | Acetone | * | Weak Inhibition of carbohydrate digesting enzymes | [41] |
| 4 | *A. ferox* | Leaves | Aqueous | * | Stimulate adiponectin secretion from 3T3-L1 cells | [42] |
| 5 | *A. ustulata* | * | * | * | * | * |
| 6 | *B. micrantha* | Bark, stem, roots | Methanol | Quercetin-3-*O*-β-d-glucoside (1→4)-α-L-rhamnoside (**1**) | The methanol extract exhibited high inhibition against glucosidase and moderate inhibition against alpha-amylase. The isolated compound exhibited (Quercetin-3-o-β-d-glucopyranoside (1→4)-α-L-rhamnoside) strong inhibition against alpha glucosidase | [87,88] |
| 7 | *B. discolor* | Leaves, roots and stems | Aqueous | * | High glucose utilisation in Chang liver, C2C12 muscle and 3T3-L1 fat cells | [43] |
| 8 | *B. elliptica* | Leaves | Aqueous | * | Moderate glucose utilization in HepG2 liver cells | [89] |

**Table 2.** *Cont.*

| S/N | Scientific Name | Plant Part Used | Solvent/Extract Used | Antidiabetic Isolated Compounds | Antidiabetic Mechanism of Action | References |
|---|---|---|---|---|---|---|
| 9 | *B. ilicifolia* | Leaves | Aqueous | * | Moderate glucose utilisation in HepG2 liver cells | [90] |
| 10 | *B. abyssinica* | Leaves, flowers, stems and roots | Aqueous | * | Strong Inhibition against alpha amylase | [91] |
| 11 | *B. frutescens* | Whole plant | Aqueous | * | High glucose utilisation in Chang liver and C2C12 muscle cells | [47] |
| 12 | *B. natalensis* | * | * | * | * | * |
| 13 | *C. sativa* | Leaves, roots | Aqueous | * | Stimulate glucose uptake in 3T3-L1 fat cells and strong inhibition of alpha-amylase | [43,92] |
| 14 | *C. edulis* | Leaves | Aqueous | * | Strong inhibition against alpha glucosidase. | [93] |
| 15 | *C. edulis* | Leaves | Dichloromethane/methanol (1:1) | * | Moderate glucose utilisation in C2C12 muscle and 3T3-L1 fat cells | [43] |
| 16 | *C. roseus* | Leaves and twig | Dichloromethane/methanol (1:1) | * | Moderate glucose utilisation in 3T3-L1 cells | [43] |
| 17 | *C. olearaceus* | * | * | * | * | * |
| 18 | *C. baccifera* | Whole plant | Aqueous | * | Moderate glucose utilisation in Chang liver, C2C12 and 3T3-L1 cells | [43] |
| 19 | *C. capensis* | Leaves | Aqueous | * | Increase glucose utilisation in 3T3-L1 cells | [43] |
| 20 | *C. scabrida* | * | * | * | * | * |
| 21 | *D. thunbergii* | Roots | Ethanol and Aqueous | * | Moderate glucose utilisation in L6 muscle cells | [53] |
| 22 | *E. natalensis* | Root-bark | Acetone | * | Strong inhibition against alpha amylase | [41] |
| 23 | *E. undulata* | Stem-bark and root-bark | Acetone | Epicatechin (**2**), α-amyrin-3O-β-(5-hydroxy) ferulic acid (**3**) | High glucose utilisation in Chang liver, C2C12 and 3T3-L1 cells. The isolated compound (α-amyrin-3O-β-(5-hydroxy) ferulic acid) exhibited strong inhibition against carbohydrate digesting enzymes | [7,56] |
| 24 | *H. gymnocomum* | * | * | * | * | * |
| 25 | *H. nudifolium* | * | * | * | * | * |
| 26 | *H. odoratissimum* | leaf and stem | Aqueous | * | Hypoglycaemic effect | [94] |
| 27 | *H. petiolare* | Whole plant | Aqueous | * | High glucose utilisation in L6 cells Strong inhibition against carbohydrate digesting enzymes | [95] |
| 28 | *H. arborescens* | Leaf | Aqueous and Ethanol | * | Moderate glucose utilization in C3A and L6 cells and strong inhibition against alpha amylase activity | [96] |
| 29 | *H. argentae* | Corms | Aqueous | * | Moderate glucose utilisation in HepG2 liver and L6 muscle cells, stimulate INS-1 cells proliferation and prevent fat accumulation in 3T3-L1 adipocytes. | [53] |
| 30 | *H. colchicifolia* | Corms | Methanol | Hypoxoside (**4**), bisphenol A diglycidyl ether (**5**) | Moderate glucose utilisation in C2C12 muscle cells and Chang liver cells | [58] |

**Table 2.** *Cont.*

| S/N | Scientific Name | Plant Part Used | Solvent/Extract Used | Antidiabetic Isolated Compounds | Antidiabetic Mechanism of Action | References |
|---|---|---|---|---|---|---|
| 31 | *H. hemerocallidea* | Leaves and bark | Acetone and Hexane | β -Sitosterol (**6**) | The extract exhibited moderate Inhibition against alpha-amylase and strong inhibition against alpha glucosidase. The isolated compound (β -sitosterol) stimulates insulin release | [97,98] |
| 32 | *L. tetragonia* | * | * | * | * | * |
| 33 | *L. leonorus* | Leaves | Aqueous | Marrubin (**7**) | Hypoglycaemic effect | [30,63] |
| 34 | *M. balsamina* | Stems and flowers | Dichloromethane/ methanol (1:1) | Betulinic acid (**8**) | High glucose utilisation in 3T3-L1 cells, Chang liver and C2C12. The isolated compound (betulinic acid) exhibited antihyperglycemic effect | [43,99] |
| 35 | *M. foetida* | Whole plant | Dichloromethane/ methanol (1:1) | Foetidin (**9**) | Moderate glucose utilisation in C2C12 muscle cells | [43] |
| 36 | *O. longibracteatum* | Bulbs | Aqueous and ethanol | * | High glucose utilisation in Chang liver and C2C12 muscle cells | [47] |
| 37 | *P. guajava* | Leaves and roots | Dichloromethane/ methanol (1:1) | lanost-7-en-3β-ol-26-oic acid (**10**) lanost-7-en-3β, 12β-diol-26-oic acid (**11**), lanost-7-en-3β, 12β, 29-triol-26-oic acid (**12**), lanost-7-en-3β-ol-26-oic acid-3β-glucoside (**13**), gallic acid (**14**), quercetin (**15**), kaempferol (**16**), myricetin (**17**), catechin (**18**) | Moderate glucose uptake in C2C12 and 3T3-L1 cells. Exhibited strong inhibition against alpha amylase | [41,43] |
| 38 | *R. graveolens* | Arial parts | Aqueous and ethanol | * | High glucose utilisation in Chang liver cells | [47] |
| 39 | *S. birrea* | Bark | Aqueous and methanol | * | High glucose utilisation in C2C12, 3T3-L1 and HepG2 cells. Strong Inhibition of carbohydrate digestive enzymes | [7] |
| 40 | *S. aculeastrum* | * | * | * | * | * |
| 41 | *S. henningsii* | Stem bark | Aqueous | * | High glucose uptake in differentiated 3T3-L1 cells and moderate inhibition against alpha glucosidase and weak inhibition against alpha amylase | [100] |
| 42 | *S. frutescens* | Leaves and shoots | Aqueous | Pinitol (**19**) | Prevention of insulin resistance in Chang liver cells, stimulate secretion of insulin from INS-1 cells and high glucose uptake in C2C12 muscle and 3T3-L1 fat cells | [73–75] |
| 43 | *T. camphoratus* | Leaves and twig | Aqueous and ethanol | * | High glucose uptake in C2C12 cells | [47] |
| 44 | *T. alliacea* | * | * | * | * | * |
| 45 | *T. violacea* | Whole plant | Aqueous and ethanol | * | High glucose utilisation in Chang liver and C2C12 cells | [47] |
| 46 | *V. amygdalina* | Leaves | Aqueous, methanol, acetone | * | High glucose uptake in C2C12 and Chang liver cells and Strong inhibition of carbohydrate digesting enzyme | [7,34] |

**Table 2.** *Cont.*

| S/N | Scientific Name | Plant Part Used | Solvent/Extract Used | Antidiabetic Isolated Compounds | Antidiabetic Mechanism of Action | References |
|---|---|---|---|---|---|---|
| 47 | *Vernonia oligocephala* Sch. Bip. | Leaves | Methanol | Oligocephalate (**20**) | The extract exhibited weak inhibition against alpha amylase. The isolated compound showed strong inhibition against alpha glucosidase | [78,79] |
| 48 | *V. major* | Leaves, roots and stems | Dichloromethane/ methanol (1:1) | * | High glucose uptake in Chang liver, C2C12 and 3T3-L1 cells | [43] |

*: No reported antidiabetic activities available in the literature.

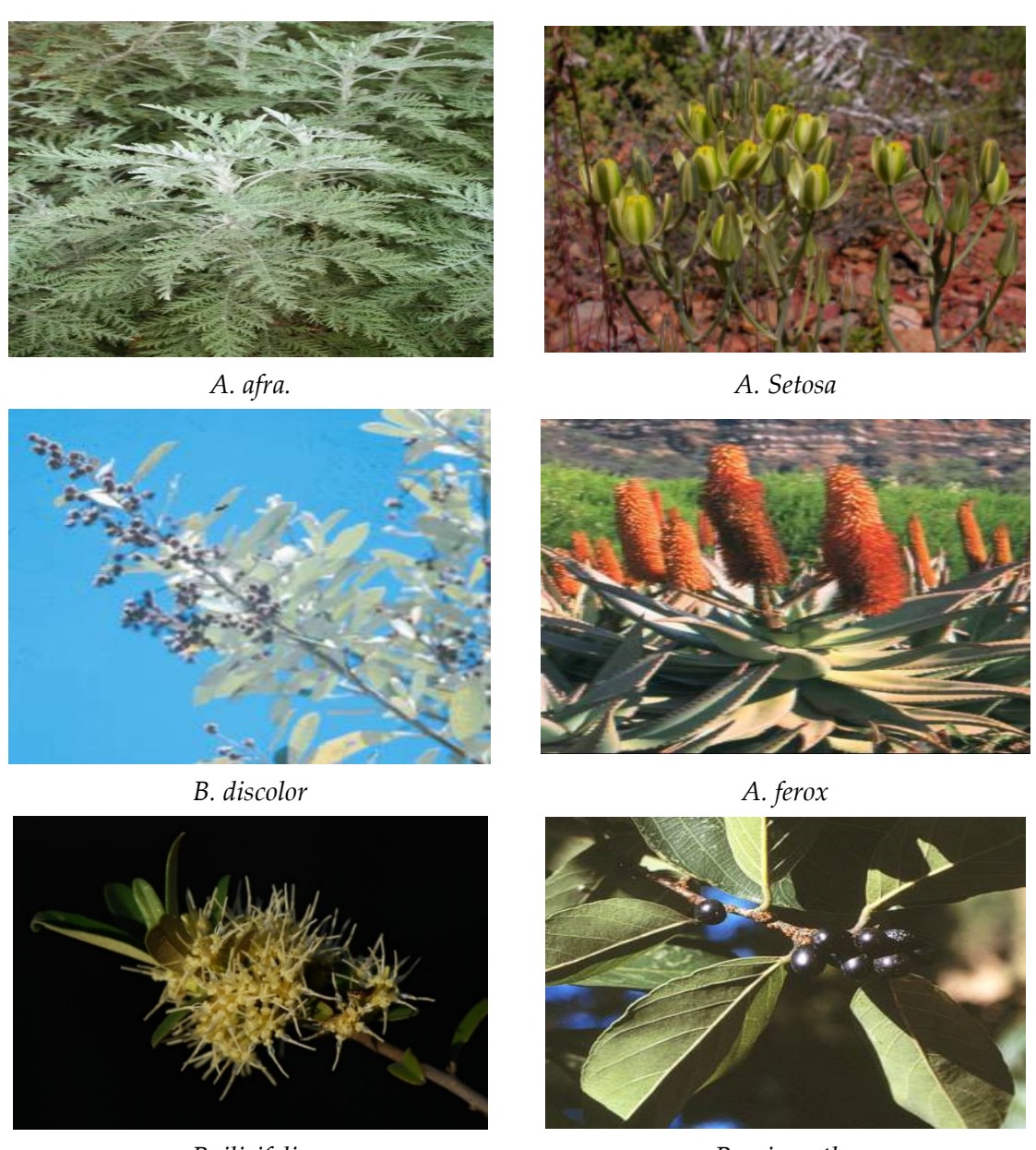

*A. afra.*

*A. Setosa*

*B. discolor*

*A. ferox*

*B. ilicifolia*

*B. micrantha*

**Figure 3.** *Cont.*

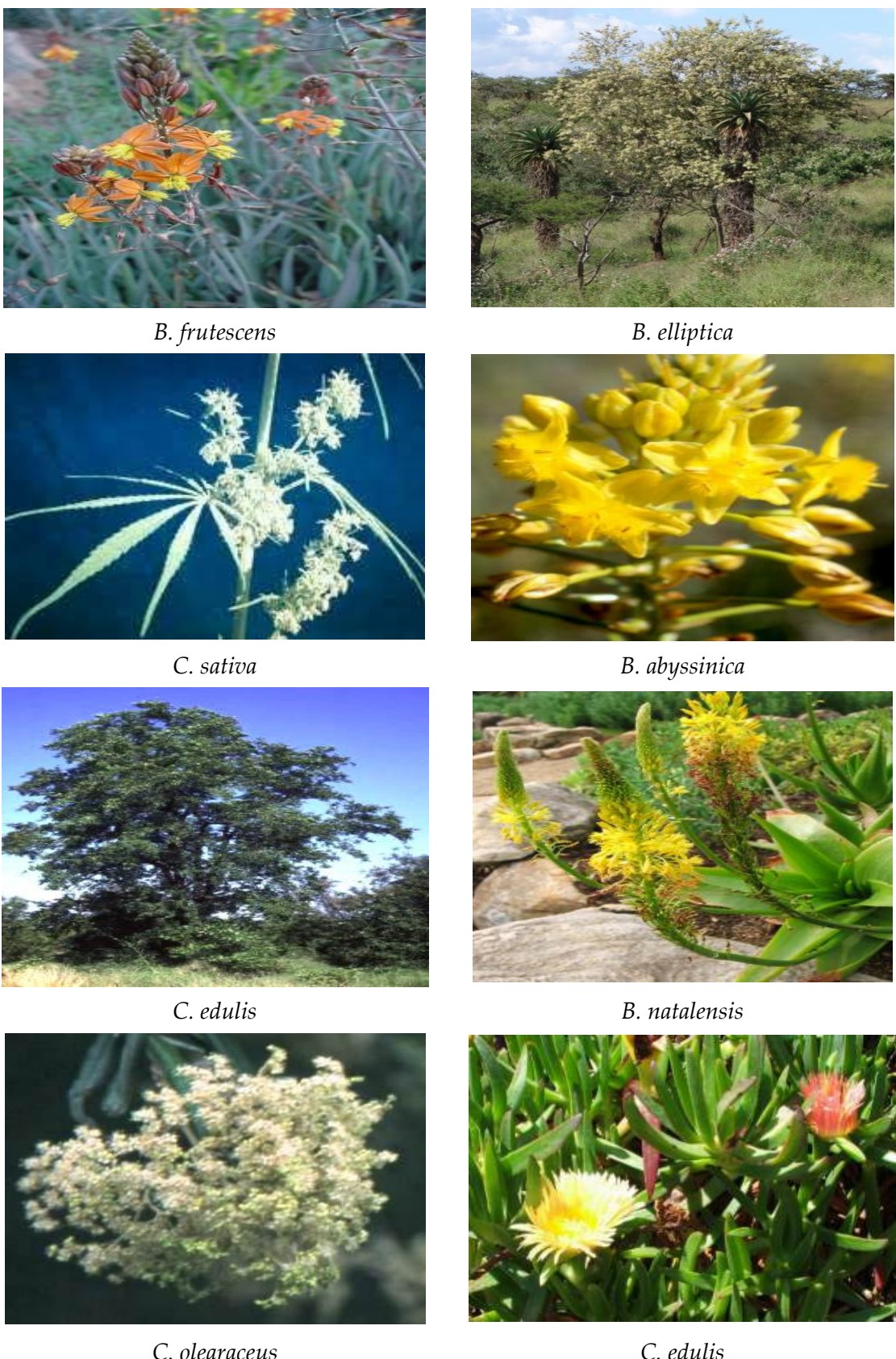

*B. frutescens*

*B. elliptica*

*C. sativa*

*B. abyssinica*

*C. edulis*

*B. natalensis*

*C. olearaceus*

*C. edulis*

**Figure 3.** *Cont.*

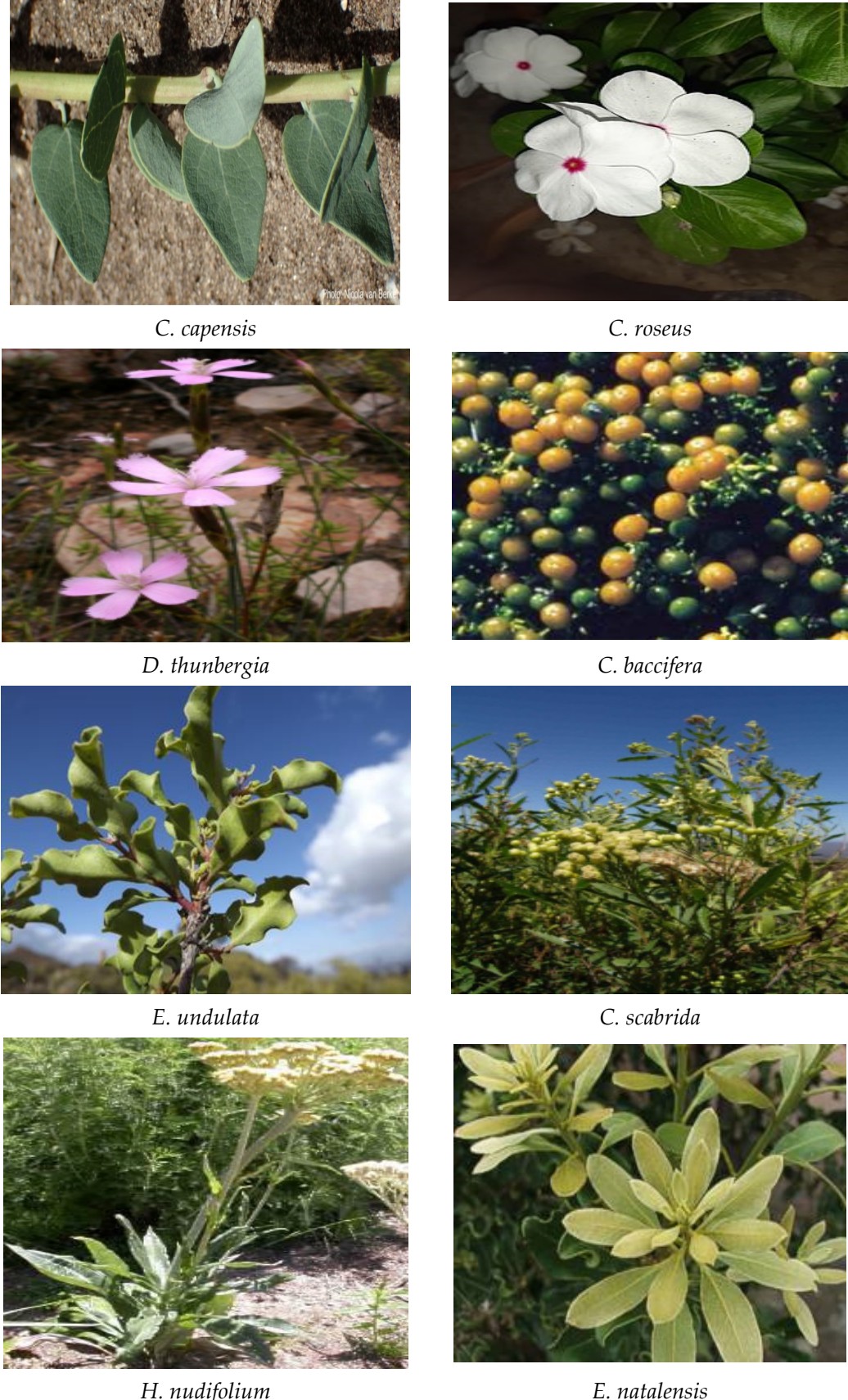

*C. capensis*

*C. roseus*

*D. thunbergia*

*C. baccifera*

*E. undulata*

*C. scabrida*

*H. nudifolium*

*E. natalensis*

**Figure 3.** *Cont.*

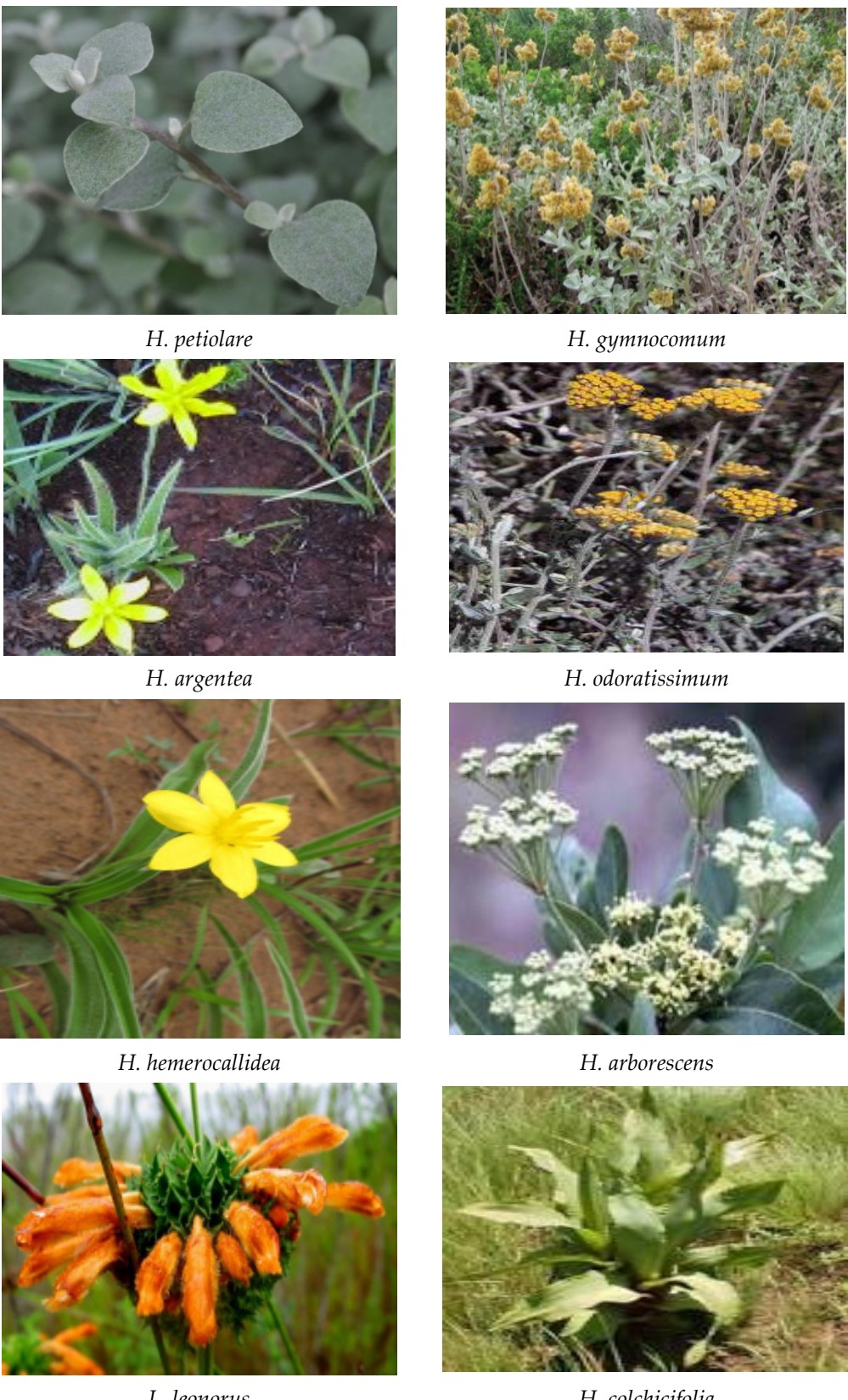

*H. petiolare*

*H. gymnocomum*

*H. argentea*

*H. odoratissimum*

*H. hemerocallidea*

*H. arborescens*

*L. leonorus*

*H. colchicifolia*

**Figure 3.** *Cont.*

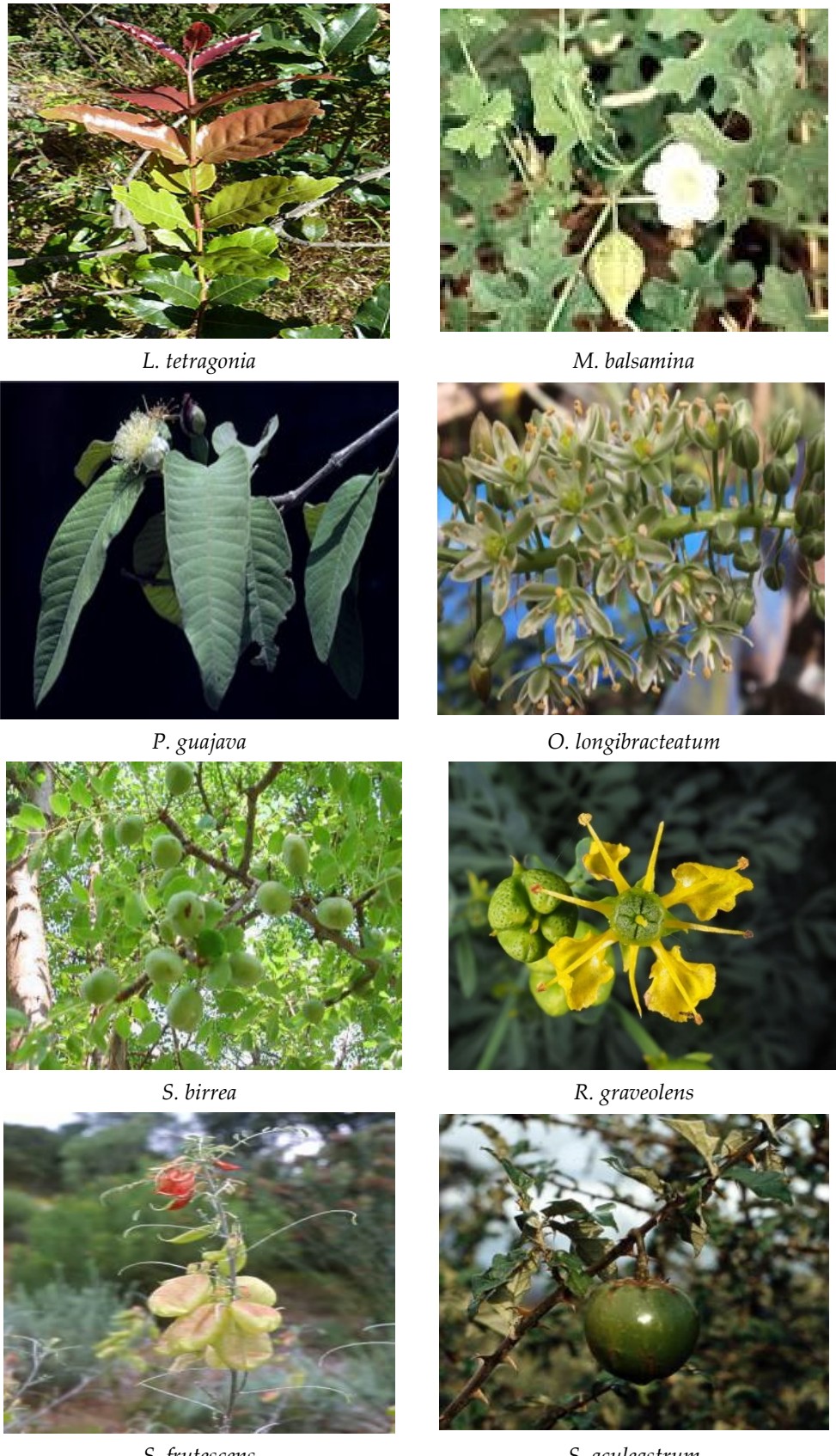

*L. tetragonia*

*M. balsamina*

*P. guajava*

*O. longibracteatum*

*S. birrea*

*R. graveolens*

*S. frutescens*

*S. aculeastrum*

**Figure 3.** *Cont.*

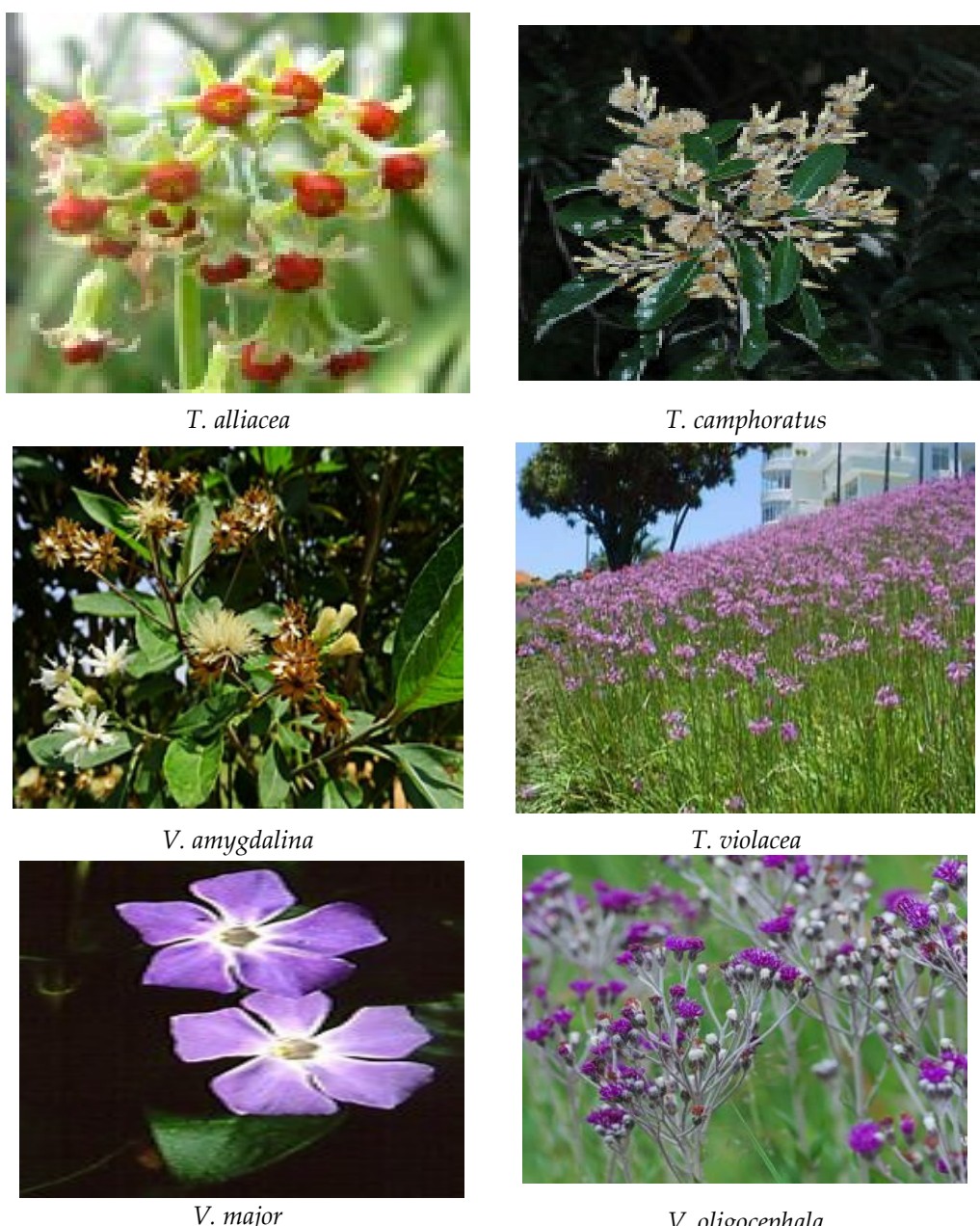

**Figure 3.** Selected medicinal plants used by the people of Eastern Cape for the treatment of diabetes [101–143].

## 4. Conclusions and Future Direction

The present trend in the management of diabetes involves the use of medicinal plants, since oral antidiabetic drugs are known to be expensive with unwanted side effects. This high prevalence justifies the special attention toward the use of these plants for the management of diabetes. In this review, it is evident that the majority of these medicinal plants exert their antidiabetic mechanism of action through the enhancement of glucose uptake in cells (hepatic, skeletal and fat cells), stimulation of the release of insulin by pancreatic beta-cells or through the alteration of certain hepatic enzymes involved in glucose metabolism and reducing intestinal glucose absorption. Out of the 48 medicinal plants used in the Eastern Cape for the treatment of diabetes, only eight plants have not been scientifically studied. Furthermore, it is imperative to note that a large number of active antidiabetic molecules of these plants are yet to be isolated and clinically studied. Thus, an effort needs to be devoted

to the isolation and purification of these antidiabetic molecules and the determination of their mechanism of action, both in in vitro and in vivo experimental animal models.

**Author Contributions:** Conceptualization: I.J.S. and A.A.H.; Writing of the original draft: I.J.S.; Editing: I.J.S. and A.A.H.; Supervision: A.A.H. All authors have read and agreed to the published version of the manuscript.

**Funding:** This research received no external funding.

**Data Availability Statement:** Not applicable.

**Acknowledgments:** The authors are thankful to the Cape Peninsula University of Technology, for publication support.

**Conflicts of Interest:** The authors state that there is no conflict of interest in publishing this article.

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
