# Peer review of "Antidiabetic Medicinal Plants Used in the Eastern Cape Province of South Africa: An Updated Review"

_processes, doi:10.3390/pr10091817_

Round 1

Reviewer 1 Report

The authors described "Antidiabetic medicinal plants used in the Eastern Cape province of South Africa: An updated review". Well I don’t think so that the title suits these words “updated review" because the article still lack many sides of studies for diabetic potential. I have find out that authors mentioned all the studies of “Extracts” of plants showing diabetic potential. I am wondered there is not work out in Cape province on some pure isolated compounds from plants showing anti-diabetic potential????. I strongly recommend authors find literature of pure isolated compounds (New/known) both ok which shows that these compounds have anti-diabetic potential. After finding out these studies I recommend to add this data in the table by addition of one new column and mention “pure compound/extract” and then in this section the authors can add if the activity was achieved by pure compound or extract.

Furthermore, I also strongly recommend authors to mention the clear anti-diabetic activity of every plants. As like in table 2 reference 37 and 41 are graduation thesis, and I found out from these thesis studies that the plants which are mentioned in these studies have weak anti-diabetic activity so authors should mention it in the table if any plants have strong, moderate or weak activity so that it can give a clear picture of every plant for future considerations.

Author Response

                                                                     CHEMISTRY DEPARTMENT

                                                                      Group of Natural Products Chemistry

                                                                         Dr. Idowu Jonas Sagbo

                                                                          Mail: [email protected]

                                                                   Tel: +27 621320933

August 30, 2022.

Processes

Submission of Revised Paper [Manuscript ID: processes-1894150]

We would like to thank you and the reviewer for a careful and thorough reading of our manuscript and for the thoughtful comments and constructive suggestions which help to improve the quality of this manuscript.

 We have revised the manuscript accordingly. Our response is given in a point-by-point below.

We hope the revised version is now suitable for publication and looking forward to hearing from you in due course.

Sincerely

Idowu Jonas Sagbo (PhD)

Reviewer 1

Reviewer’s comments

Original comments of the reviewer:

  1. The authors described "Antidiabetic medicinal plants used in the Eastern Cape province of South Africa: An updated review". Well I don’t think so that the title suits these words “updated review" because the article still lack many sides of studies for diabetic potential. I have find out that authors mentioned all the studies of “Extracts” of plants showing diabetic potential. I am wondered there is not work out in Cape province on some pure isolated compounds from plants showing anti-diabetic potential????. I strongly recommend authors find literature of pure isolated compounds (New/known) both ok which shows that these compounds have anti-diabetic potential. After finding out these studies I recommend to add this data in the table by addition of one new column and mention “pure compound/extract” and then in this section the authors can add if the activity was achieved by pure compound or extract.

Furthermore, I also strongly recommend authors to mention the clear anti-diabetic activity of every plants. As like in table 2 reference 37 and 41 are graduation thesis, and I found out from these thesis studies that the plants which are mentioned in these studies have weak anti-diabetic activity so authors should mention it in the table if any plants have strong, moderate or weak activity so that it can give a clear picture of every plant for future considerations.

Reply by the author(s):

We thank you for these comments. We have now reported the antidiabetic isolated compounds from the Eastern Cape plants, and we have added these antidiabetic isolated compounds together with their antidiabetic activities in the table as suggested.  In addition, we have also drawn the structures of these compounds for better understanding. Please see the manuscript for confirmation.

We have mentioned the clear anti-diabetic activity of every plants as suggested. We have also indicated if any plant has strong, moderate, or weak activity.  Please see the manuscript for confirmation.

Reviewer 2 Report

REVIEW

“Antidiabetic medicinal plants used in the Eastern Cape province of South Africa: An updated review”

Recommendation: The manuscript claims an improvement.

Please take care of the following points before resubmission:

·         The manuscript is poorly written and the standard of English is low, frequently making it hard to understand the authors meaning and what they actually did. The mistakes in English are too numerous to list here. Language needs extensive improvement.

·         Please use the correct format of the references required for the journal (check thoroughly).

·         Scientific name must be written with full author’s citations. Here you must write in this way: Galega officinalis L. ; Albuca setosa Jacq. ; etc.

·         The article has been just summarised the works of different scientists. There is no indication of proper outcomes of this review work. It would be best if you highlighted the novelty of this study.

·         No methodology is mentioned here. The PRISMA guidelines are not followed. Authors must provide a systematic methodology in their revised manuscript.

·         The diagrams are not up to the mark of journal standard. It needs improvement to reach the journal standard.

·         Conclusion: Make it more precise and focus on the relevance of the present study. Eliminate the irrelevant topics, which are defocusing the primary objective of this article. Rewrite this.

·         Abstract looks like an introduction paragraph. Your abstract must be brief, precise and meaningful.

Author Response

                                                                     CHEMISTRY DEPARTMENT

                                                                      Group of Natural Products Chemistry

                                                                         Dr. Idowu Jonas Sagbo

                                                                          Mail: [email protected]

                                                                         Tel: +27 621320933

August 30, 2022.

Processes

Submission of Revised Paper [Manuscript ID: processes-1894150]

We would like to thank you and the reviewer for a careful and thorough reading of our manuscript and for the thoughtful comments and constructive suggestions which help to improve the quality of this manuscript.

 We have revised the manuscript accordingly. Our response is given in a point-by-point below.

We hope the revised version is now suitable for publication and looking forward to hearing from you in due course.

Sincerely

Idowu Jonas Sagbo (PhD)

Reviewer 2

Original comments of the reviewer:

  1. The manuscript is poorly written and the standard of English is low, frequently making it hard to understand the authors meaning and what they actually did. The mistakes in English are too numerous to list here. Language needs extensive improvement.

Reply by the author(s):

 We thank you. We have revised the entire manuscript for English editing as suggested. Please check the manuscript.

Original comments of the reviewer:

  1. Please use the correct format of the references required for the journal (check thoroughly).

Reply by the author(s):

We thank you for this comment. We have now checked and used the correct format of the references required for the journals.

Original comments of the reviewer:

Scientific name must be written with full author’s citations. Here you must write in this way: Galega officinalis L. ; Albuca setosa Jacq. ; etc.

Reply by the author(s):

We thank you for this comment. We agree that the scientific name should be written in full author’s citation, and we have written all the plants scientific names in full author’s citation as suggested.  It is imperative to note that after the scientific name has been written in full for the first time in the manuscript, it can now be abbreviated in the entire manuscript. For example, “Galega officinalis L” for the first time. It can now be abbreviated subsequently in the entire manuscript as “G. officinalis”.

Original comments of the reviewer:

The article has been just summarised the works of different scientists. There is no indication of proper outcomes of this review work. It would be best if you highlighted the novelty of this study.

Reply by the author(s):

We thank you for this comment. It is imperative to note that this paper is a review study. A review article or study is an article/study that summarizes the current state of understanding on a topic within a certain discipline. The present study focused on a comprehensive review of antidiabetic medicinal plants used in the Eastern Cape province of South Africa in the management of diabetes mellitus to help researchers and government agencies to prevent the possible extinction of these antidiabetic plants and also provides guidance for future research on the scientifically under-exploited antidiabetic medicinal plants and their active molecules for drug discovery and development. In addition, we also proposed that the target tissue (s) and mechanism (s) of action of exploited and unexploited plants deserve further investigations. It should be noted that this is the first study on assessments of the therapeutic profiles of the Eastern Cape antidiabetic medicinal plants and their active molecules for the management of diabetes mellitus. Furthermore, many of these antidiabetic medicinal plants are very new.

Original comments of the reviewer:

No methodology is mentioned here. The PRISMA guidelines are not followed. Authors must provide a systematic methodology in their revised manuscript.

Reply by the author(s):

We thank you for this comment. The methodology used is included and this methodology has been previously used by several researchers in several review articles. Please see the following published articles for confirmation:

  1. Akinyede, K.A.; Cupido, C.N.; Hughes, G.D.; Oguntibeju, O.O.; Ekpo, O.E. Medicinal Properties and In Vitro Biological Activities of Selected Helichrysum Species from South Africa: A Review. Plants 2021, 10, 1566. https://doi.org/10.3390/ plants10081566.

  1. Odukoya, J.O.; Odukoya, J.O.; Mmutlane, E.M.; Ndinteh, D.T. Ethnopharmacological Study of Medicinal Plants Used for the Treatment of Cardiovascular Diseases and Their Associated Risk Factors in sub-Saharan Africa. Plants 2022, 11, 1387. https://doi.org/10.3390/ plants11101387

Original comments of the reviewer:

The diagrams are not up to the mark of journal standard. It needs improvement to reach the journal standard.

Reply by the author(s):

We thank you for this comment. We have now improved the diagram as suggested.

Original comments of the reviewer:

Conclusion: Make it more precise and focus on the relevance of the present study. Eliminate the irrelevant topics, which are defocusing the primary objective of this article. Rewrite this.

Reply by the author(s):

We thank you for this comment. We have now eliminated irrelevant topics. we have also rewritten the conclusion as suggested.

Original comments of the reviewer:

Abstract looks like an introduction paragraph. Your abstract must be brief, precise an meaningful.

Reply by the author(s):

We thank you for this comment. We have now made the abstract brief and precise as suggested.

Round 2

Reviewer 1 Report

The authors addressed all the questions very clear. I recommend for its publication.